# Targeted, homology-driven gene insertion in stem cells by ZFN-loaded 'all-in-one' lentiviral vectors

Yujia Cai[1], Anders Laustsen[1], Yan Zhou[1], Chenglong Sun[1], Mads Valdemar Anderson[1,2], Shengting Li[1,3], Niels Uldbjerg[4], Yonglun Luo[1], Martin R Jakobsen[1], Jacob Giehm Mikkelsen[1]*

[1]Department of Biomedicine, Aarhus University, Aarhus, Denmark; [2]Technical University of Denmark, Lyngby, Denmark; [3]Beijing Genomics Institute, Shenzhen, China; [4]Department of Clinical Medicine, Aarhus University, Aarhus, Denmark

**Abstract** Biased integration remains a key challenge for gene therapy based on lentiviral vector technologies. Engineering of next-generation lentiviral vectors targeting safe genomic harbors for insertion is therefore of high relevance. In a previous paper (*Cai et al., 2014a*), we showed the use of integrase-defective lentiviral vectors (IDLVs) as carriers of complete gene repair kits consisting of zinc-finger nuclease (ZFN) proteins and repair sequences, allowing gene correction by homologous recombination (HR). Here, we follow this strategy to engineer ZFN-loaded IDLVs that insert transgenes by a homology-driven mechanism into safe loci. This insertion mechanism is driven by time-restricted exposure of treated cells to ZFNs. We show targeted gene integration in human stem cells, including CD34[+] hematopoietic progenitors and induced pluripotent stem cells (iPSCs). Notably, targeted insertions are identified in 89% of transduced iPSCs. Our findings demonstrate the applicability of nuclease-loaded 'all-in-one' IDLVs for site-directed gene insertion in stem cell-based gene therapies.

*For correspondence: giehm@biomed.au.dk

Competing interests: The authors declare that no competing interests exist.

## Introduction

Uncontrolled genomic insertion of transgenes, leading to insertional mutagenesis, remains a primary challenge for safe genetic therapies. Lentiviral vectors offer an effective gene delivery route, but also suffer from prevalent gene insertion into transcriptionally active regions (*Schröder et al., 2002*; *Modlich et al., 2009*). Novel strategies to attain targeted gene integration (TGI) into predetermined genomic loci are highly warranted. TGI using lentiviral vectors can be achieved by delivering donor sequences for insertion by homologous recombination (HR) into double-strand breaks (DSBs) generated by tailored endonucleases (*Lombardo et al., 2011*). Development of programmable nucleases, ZFNs, TALENs, and now CRISPR/Cas9, has made effective gene disruption feasible, even in stem cells, but genetic operations involving TGI in stem cells are still challenged by low membrane permeability, low HR efficiency, and high sensitivity to external manipulation (*Genovese et al., 2014*; *Hockemeyer et al., 2011*; *Hendel et al., 2015*; *Tay et al., 2013*; *Tiyaboonchai et al., 2014*).

We previously established genomic editing in cells treated with 'all-in-one' lentiviral particles loaded with locus-targeted ZFN proteins and a donor sequence flanked by homology arms (*Cai et al., 2014a*). Here, building on these findings, we report a simple method for establishing TGI by treating cells with integrase-defective lentiviral vectors (IDLVs) packaged with both the transgene-containing donor template and ZFN proteins fused to the N-terminus of lentiviral Gag and GagPol polypeptides. Locus-directed cleavage and TGI is evident in cell lines, CD34[+] hematopoietic stem cells, and induced pluripotent stem cells (iPSCs) upon transduction with ZFN-loaded vectors.

Notably, the prevalence of TGI in two safe harbors in the human genome, the *CCR5* and *AAVS1* loci, is high in iPSCs treated with ZFN-loaded lentiviral vectors, resulting in non-mosaic clones harboring a site-directed gene insertion and no additional cutting at top-ranked off-target sites.

## Results

IDLVs harboring both ZFN protein and vector RNA with the transgene flanked by homology arms are here designated 'IDLV-ZFN(locus)/*gene*', where '*gene*' refers to the transgene cassette flanked by homology arms and transferred by the vector. Principles of production, transduction, and integration of such vectors are illustrated in *Figure 1A*. Together with a dimer of lentiviral vector RNAs, approximately 5000 copies of Gag and GagPol polypeptides (of which about half carry a ZFN fused to the matrix protein) form the capsid, which is wrapped in a segment of the membrane during budding from the producer cells. ZFNs are released from Gag and GagPol upon cleavage by the viral protease during maturation of the virus particle. The mature virus particles contain two types of ZFNs, left and right, which upon DNA binding in the target cell form a catalytically active dimer. After viral entry in the target cells and endosomal escape, vector RNA is reverse transcribed to DNA, which can serve as donor for homology-directed insertion at targeted breakage sites generated by the co-delivered ZFNs (*Figure 1A*, lower panel).

To visualize the enrichment of ZFN-containing lentiviral polypeptides at the membrane of the producer cells, we transfected HEK293T cells with GagPol plasmid harboring either the coding sequence for a HA-tagged ZFN (HA-ZFN(gfp)) recognizing the e*gfp* gene or the *egfp* gene itself fused to the 5'-end of *gag* gene and analyzed the virus-producing cells using confocal microscopy (*Figure 1B* and *Figure 1—figure supplement 1A*). By staining with an antibody specific for the tag, we found that the ZFNs were highly enriched at the cell membrane (*Figure 1B*). We then visualized the lentiviral particles (designated LPs since the vector was not included) and found for both LP-HA-ZFN(gfp) and LP-eGFP co-localization of the fusion protein with viral p24 protein (*Figure 1C* and *Figure 1—figure supplement 1B and C*), suggesting that ZFN (or eGFP) proteins were indeed packaged into the LPs. To measure the lifespan of LP-delivered ZFNs in transduced cells, HEK293 cells were synchronized at 4°C for 60 min before fixation at different time points (1, 12, 24, and 48 hr, respectively) after transduction with ZFN-loaded LPs. Already within the first hour after exposure to LP-HA-ZFN(gfp), the ZFNs were easily detectable inside the cells, whereas after 24 hr essentially all ZFNs had been degraded or diluted (*Figure 1D*). These findings showed that nucleases delivered by LPs were immediately available after transduction and that the time of action was restricted due to decay of the proteins.

We then performed proof-of-efficacy studies by transducing HEK293T cells with IDLVs (500 ng p24), with or without *AAVS1*-directed ZFNs, carrying an eGFP expression cassette (IDLV-ZFN (AAVS1)/*egfp*). Initially, three days post-transduction, a higher frequency of eGFP$^+$ cells was observed among cells transduced with IDLV/*egfp* than for cells transduced with IDLV-ZFN(AAVS1)/*egfp* (59.1% versus 44.7%), indicating that the overall transduction potential was higher for IDLVs that did not carry ZFNs (*Figure 1E* and *Figure 1—figure supplement 1D and 1E*). Nevertheless, at later time points 2.5% of the cells were eGFP$^+$ after treatment with IDLV-ZFN(AAVS1)/*egfp*, whereas the percentage dropped to below 1% in IDLV/*egfp*-treated cells (*Figure 1E* and *Figure 1—figure supplement 1E*). PCR amplification and sequencing of the 5' and 3' junction sites including part of the donor sequence and the regions flanking the inserted transgene cassette showed that events of correct integration into *AAVS1* locus were evident after treatment with IDLV-ZFN(AAVS1)/*egfp*, whereas PCR products indicative of site-directed insertion did not appear for IDLV/*egfp* (*Figure 1F*). However, DSBs were not repaired by HR only, and mismatches introduced by non-homologous end joining (NHEJ) were detected in 5% of the *AAVS1* alleles, as measured by Surveyor nuclease assay (*Figure 1G*). To evaluate the levels of TGI with another reporter gene, we treated HEK293T cells with IDLV-ZFN(AAVS1)/*fluc* carrying *AAVS1*-targeted ZFNs and a *firefly luciferase (fluc)* expression cassette. Indeed, *fluc* gene expression was maintained for up to 18 days after transduction with IDLV-ZFN(AAVS1)/*fluc* (*Figure 1H*), although the levels dropped from the initial 3-day time point, where episomal forms were still likely to support expression. For IDLV/*fluc*, in contrast, luminescence dropped to almost undetectable levels. TGI of the *fluc* gene into the *AAVS1* locus was verified by PCR (*Figure 1I*), and the rate of NHEJ estimated to be 8% (*Figure 1J*).

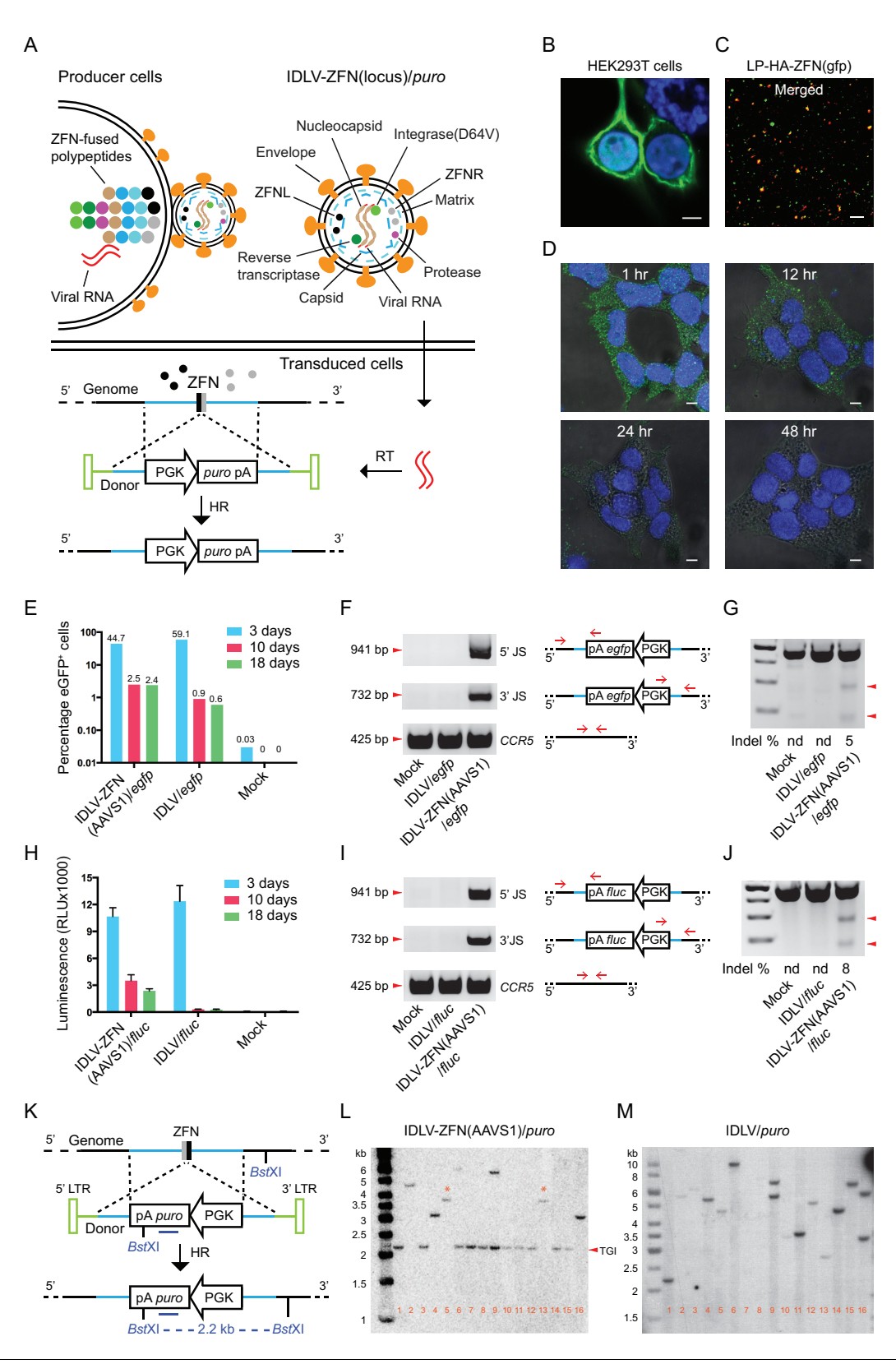

**Figure 1.** Proof-of-efficacy studies and characterization of ZFN-loaded IDLVs carrying a transgene flanked by homology arms facilitating insertion by homologous recombination. (A) Schematic illustration of the production, transduction and integration of IDLVs carrying ZFNs and transgene.

*Figure 1 continued on next page*

*Figure 1 continued*

Enrichment of ZFN-fused viral polypeptides and donor-carried viral RNAs at the cell membrane leads to assembly of ZFN-containing virions. During virus maturation, the polypeptides are cleaved into enzymatic and structural proteins by HIV protease. In the transduced cells, the viral RNAs are reverse transcribed to DNA that serves as donor for repair of ZFN-induced DSB by HR, leading to site-specific insertion of a gene expression cassette (here containing the *puro* gene). RT, reverse transcription; HR, homologous recombination. (**B–D**) Immunostaining and confocal microscopy of producer cells, virions, and transduced cells. Green, HA-tagged ZFN-GagPol or ZFN; blue, nuclei; red, p24; scale bar indicates 5 μm. (**B**) HEK293T cells co-transfected by pMD.2G, pHA-ZFNL(gfp)-PH-gagpol-D64V and pHA-ZFNR(gfp)-PH-gagpol-D64V. (**C**) Incorporation of ZFNs into lentiviral particles was confirmed by co-localization of HA-ZFNs and p24 (see also ***Figure 1—figure supplement 1B***). (**D**) Recording of the ZFN lifespan in HEK293 cells treated with LP-HA-ZFN(gfp) by immunostaining at different time points after treatment. (**E–G**) TGI by IDLV-ZFN(AAVS1)/*egfp* in HEK293T cells. (**E**) IDLV-ZFN(AAVS1)/*egfp* transduction resulted in higher percentage of eGFP⁺ cells relative to the IDLV/*egfp* counterpart after 10 and 18 days. (**F**) PCR analyses of TGI at the *AAVS1* locus. Illustration shows primer designs with one primer matching inside the insert and one in the flanking region allowing amplification of 5′ and 3′ junction sites (5′ and 3′ JS). Amplification of the *CCR5* locus included as a control. (**G**) Rate of imprecise gene repair by NHEJ in target cells. (**H-J**) TGI by IDLV-ZFN(AAVS1)/*fluc* in HEK293T cells. (**H**) Higher levels of Fluc expression in IDLV-ZFN(AAVS1)/*fluc*-treated HEK293T cells relative to IDLV/*fluc*-treated cells 10 and 18 days after transduction. (**I**) PCR analyses of TGI at the *AAVS1* locus by IDLV-ZFN(AAVS1)/*fluc*. (**J**) Rates of imprecise NHEJ in target cells induced by IDLV-ZFN(AAVS1)/*fluc*. (**K**) Schematic representation of Southern blot for analysis of TGI at *AAVS1* locus using IDLV-ZFN(AAVS1)/*puro*. The probe used is indicated by the dark blue line; cleavage site for the used restriction enzyme (*Bst*XI) is also indicated. (**L–M**) Southern blot analyses of puromycin-resistant HEK293T clones resulting from treatment with IDLV-ZFN(AAVS1)/*puro* (**L**) and IDLV/*puro* (**M**). Fragments indicated with * are results of NHEJ leading to imprecise transgene insertion in the *AAVS1* locus.

The following figure supplement is available for figure 1:

**Figure supplement 1.** Characterization of protein delivery by ZFN-loaded IDLVs.

Next, we created ZFN-loaded IDLVs carrying the puromycin resistance gene (IDLV-ZFN(AAVS1)/*puro*) allowing us to easily isolate single puromycin-resistant clones and further scrutinize *AAVS1*-targeted integration events by Southern blot analysis (***Figure 1K***). Among 16 clones generated with IDLV-ZFN(AAVS1)/*puro*, a 2.2-kb fragment, indicative of precise TGI by HR, was detected in 11 clones (***Figure 1L***). Other types of integrations, resulting in lower-mobility bands, were evident in the five remaining clones. Notably, two clones (clone 5 and clone 13) showed a 3.7-kb band, and these clones therefore were among clones that were analyzed by Southern blotting using another restriction enzyme and an *AAVS1*-specific probe (***Figure 1—figure supplement 1F and 1G***). A common 6.4-kb band was observed for clones 5 and 13 (***Figure 1—figure supplement 1G***), suggesting that the full-length lentiviral DNA in these two cases was captured and inserted by chance into the ZFN-induced DSB at the *AAVS1* locus. These findings demonstrated that site-directed gene insertion into ZFN-generated DSBs also occurred, although less frequently, in an HR-independent fashion.

We analyzed 15 additional puromycin-resistant clones by PCR and found that 13 out of these clones yielded PCR products indicative of TGI (***Figure 1—figure supplement 1H***). In total, 84% (26/31) of the analyzed clones contained a targeted insertion of the *puro* gene in the *AAVS1* locus. In contrast, the fragments detected by Southern blot analysis of IDLV/*puro*-treated clones had variable sizes (***Figure 1M***), indicating that the integrations were random events that occurred at different genomic locations.

To demonstrate the feasibility of this approach in stem cells, we isolated CD34⁺ hematopoietic stem cells from human cord blood and focused initially on optimizing lentiviral transduction rates for optimal ZFN delivery. We first verified that ZFN-loaded VSV-G-pseudotyped LPs loaded with *AAVS1*-directed ZFNs resulted in targeted cleavage upon transduction of CD34⁺ cells. Hence, indels introduced through repair by NHEJ were found in 5% of the alleles (***Figure 2—figure supplement 1A***). Next, we compared the capacity of recombinant human fibronectin (RetroNectin) and EF–C peptides, the latter recently discovered artificial peptide nanofibrils, to support virus entry to CD34⁺ cells (*Yolamanova et al., 2013*; *Lump et al., 2015*). In our setting, the presence of EF-C allowed transduction of 85.1% of the cells with a standard eGFP-encoding lentiviral vector, whereas between 27.5 and 30.8% of the cells were transduced in the presence of RetroNectin (***Figure 2—figure supplement 1B***). Therefore, we chose to include EF-C for targeted integration in CD34⁺ cells.

In the first of two protocols (***Figure 2A***, orange labeling), we transduced CD34⁺ cells derived from two different donors with IDLV-ZFN(AAVS1)/*egfp* or IDLV/*egfp* (MOI = 5) and analyzed for eGFP⁺ cells 9 days post-transduction (***Figure 2B***). On average, treatment with IDLV-ZFN(AAVS1)/*egfp* resulted in (0.49 ± 0.14)% eGFP⁺ cells, which was significantly higher than the percentage of

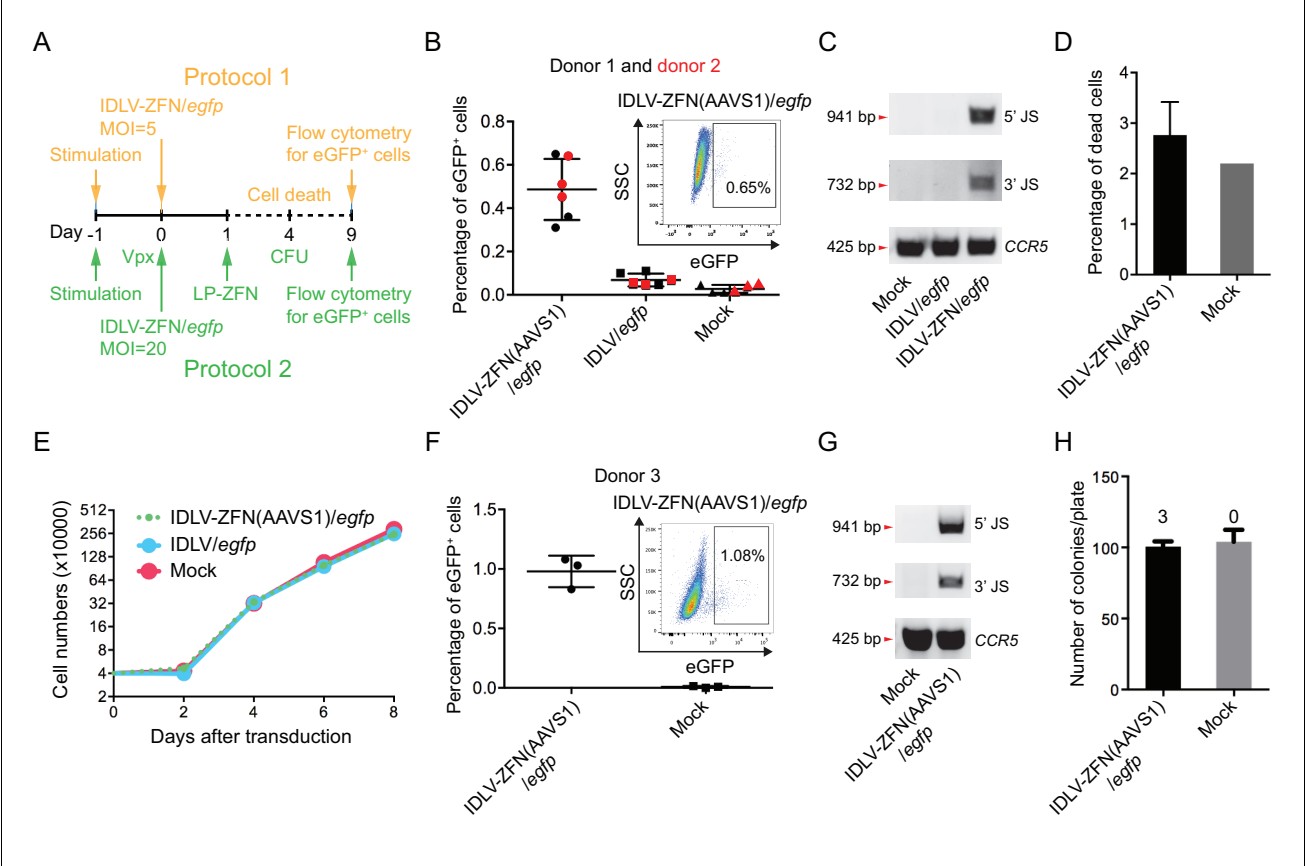

**Figure 2.** Targeted gene integration mediated by IDLV-ZFN(AAVS1)/*egfp* in CD34+ cells. (A) Schematic illustration of protocols used for IDLV-ZFN (AAVS1)/*egfp* transduction in CD34+ cells. (B) Flow cytometry analysis showing the percentage of eGFP+ cells after transduction using protocol 1 (*Figure 2A*, orange labeling). Experiments were performed in triplicates using CD34+ cells from two different donors (donor 1 and donor 2 marked by black and red color, respectively). Representative dot plot is shown on the top right. (C) PCR analyses of targeted gene integration. The primer sets used were same as in *Figure 1F*. (D) Cell death was measured 4 days after transduction. IDLV-ZFN(AAVS1)/*egfp* treatment was performed in triplicates. (E) Growth curve of transduced cells and non-transduced cells. (F) Flow cytometry analysis showing the percentage of eGFP+ cells after transduction using protocol 2 (*Figure 2A*, green labeling). Experiments were performed in triplicates using CD34+ cells from donor 3. Representative dot plot is shown on the top right. (G) PCR analyses of targeted gene integration. The primer sets used were the same as in *Figure 1F*. (H) The number of colonies formed from IDLV-ZFN(AAVS1)/*egfp*-transduced and non-transduced CD34+ cells. The experiments were performed in triplicates, and the numbers of eGFP+ colonies are provided above each bar.

The following figure supplement is available for figure 2:

**Figure supplement 1.** Allelic disruption and gene insertion after lentiviral ZFN delivery in human CD34+ cells.

eGFP+ cells (0.07 ± 0.03)% measured after transduction with the IDLV/*egfp* control without ZFNs (*Figure 2B*). PCR analyses confirmed events of TGI by IDLV-ZFN(AAVS1)/*egfp*, but not by IDLV/*egfp* (*Figure 2C*). Notably, IDLV-ZFN(AAVS1)/*egfp* transduction only slightly affected the cell viability (*Figure 2D*), and cell proliferation was not disturbed by the virus treatment (*Figure 2E*). Encouraged by this, we reasoned that the efficiency could be further increased (i) by using higher MOIs, (ii) by pretreating CD34+ cells with Vpx-loaded lentiviral particles (to degrade the reverse transcription inhibitor SAMHD1 [*Laguette et al., 2011*]), and (iii) by treating the cells with an extra dose of ZFNs delivered by LP-ZFNs. Indeed, using this improved protocol 2 (*Figure 2A*, green labeling), we were able to increase the percentage of eGFP+ cells to (0.98 ± 0.13)% after transduction with IDLV-ZFN (AAVS1)/*egfp* (*Figure 2F*). Also based on this protocol, TGI in the bulk was confirmed by PCR analysis (*Figure 2G*). Notably, IDLV-ZFN(AAVS1)/*egfp* treatment did not compromise the colony-forming capacity of CD34+ cells (*Figure 2H*). In accordance with the flow cytometry data, three eGFP+

colonies (example shown in *Figure 2—figure supplement 1C*) were identified out of a total of 302 analyzed colonies.

Ways of engineering cells by TGI may be of particular interest for therapeutic modification and use of iPSCs. To establish TGI in iPSCs using lentiviral protein transduction, we first evaluated the permissiveness of iPSCs to LPs loaded with ZFNs targeting the *AAVS1* and *CCR5* loci, respectively. LP-ZFN(AAVS1) and LP-ZFN(CCR5) disrupted 4% and 8%, respectively, of the targeted alleles in iPSCs (*Figure 3A*).

We then transduced iPSCs with IDLV-ZFN(AAVS1)/*puro*. Of the approximately 500 puromycin-resistant clones formed from 50,000 transduced iPSCs (*Figure 3B*), we initially randomly picked, isolated and expanded eight clones (clones 1–8) for further analysis. From a subsequent experiment we picked 15 additional clones (clones 9–23). PCR amplification of the 5' and 3' junction sites showed that TGI had occurred in 19 out of the 23 clones (*Figure 3C and D*). Findings based on Southern blot analysis of clones 9–23 were consistent with the PCR analysis and confirmed that 11 of these 15 clones contained an *AAVS1*-directed insertion (*Figure 3D*). Eight out of the 11 clones harbored a precise, homology-directed transgene insertion, whereas only one of the two junction sites in the remaining 3 clones (clones 9, 17, and 18) was intact. Hence, the 5' junction site was intact in clone 17, whereas clones 9 and 18 contained an intact 3' junction site, indicating that site-directed insertion in these cases involved both HR and NHEJ. Sequence analysis of the 5' and 3' junction sites amplified from clones 1–8 showed junctions sequences indicative of precise, homology-driven insertion (*Figure 3E* and *Supplementary file 1A*). For one particular clone, clone 7, we were not able to amplify the 469-bp *AAVS1* fragment, suggesting that both alleles in this clone contained a TGI (*Figure 3C*). Interestingly, sequencing of the *AAVS1* locus provided evidence of biallelic cleavage in at least one of the clones (clone 5), resulting in a 1-bp insertion in the allele that did not contain the transgene (*Supplementary file 1A*). Stemness of TGI-modified iPSCs was verified by characterization of a single representative clone using alkaline phosphatase (AP) and the TRA-1-60 pluripotency marker (*Figure 3F*). Also, genomic integrity was confirmed by karyotyping (*Figure 3G*).

To detect potential off-target cleavage by the virus-delivered ZFNs, we first sequenced the three top-ranked off-target sites in clones 1–8, but did not identify indels in any of these loci (*Supplementary file 1B*). We then isolated genomic DNA from a pool of more than 100 puromycin-resistant iPSC clones and confirmed by PCR analysis the presence of TGI events in the pool (*Figure 3H*, insert). Then, next-generation sequencing was performed on PCR amplicons representing the on-target *AAVS1* locus as well as seven top-ranked off-target sites. Among this pool of clones, most of which carry an *AAVS1*-directed gene insertion, we were not able to detect indels at off-target positions, whereas 3% of the *AAVS1* alleles that were not already targeted by transgene insertion contained indels (*Figure 3H* and *Figure 3—figure supplement 1A*). Altogether, our findings demonstrated high levels of ZFN specificity and precise TGI in the *AAVS1* locus in iPSCs treated with IDLV-ZFN(AAVS1)/*puro* vectors, resulting in at least one TGI event in 19 of 23 analyzed iPSC clones.

To demonstrate that such features of the 'all-in-one' gene delivery system was not limited to insertion into the *AAVS1* locus, we produced IDLVs (designated IDLV-ZFN(CCR5)/*puro*) carrying ZFNs targeting the human *CCR5* locus as well as vector RNA carrying the PGK-*puro* cassette flanked by homology arms matching the *CCR5* locus. Transduction of iPSCs with IDLV-ZFN(CCR5)/*puro* resulted in effective colony formation (*Figure 3I*). Among the puromycin-resistant clones, we randomly picked, isolated and expanded 15 clones (clones 1–15) and showed by PCR analysis of each of these colonies that the transgene cassette in all 15 clones (100%) had been inserted site-specifically into the *CCR5* locus (*Figure 3J*). For a single representative clone, stemness was verified using alkaline phosphatase (AP) and the TRA-1-60 pluripotency marker (*Figure 3K*). Once again, we pooled more than 100 puromycin-resistant iPSC clones and performed next-generation sequencing on PCR amplicons amplified from the on-target locus and nine high-ranked off-target sites in the pooled DNA sample (*Figure 3L* and *Figure 3—figure supplement 1B*). Notably, we identified an indel in approximately 19% of the *CCR5* alleles that did not already carry a TGI, demonstrating that the second *CCR5* allele was cleaved by ZFNs in approximately one of every five clones harboring a *CCR5*-directed insertion. Importantly, off-target cleavage and indel formation by NHEJ was not evident in any of the nine analyzed off-target candidate loci. Hence, not even the *CCR2* locus (off-target 3), which is highly homologous to the *CCR5* site and a known off-target cleavage site for *CCR5*-targeted ZFNs, harbored detectable levels of indels.

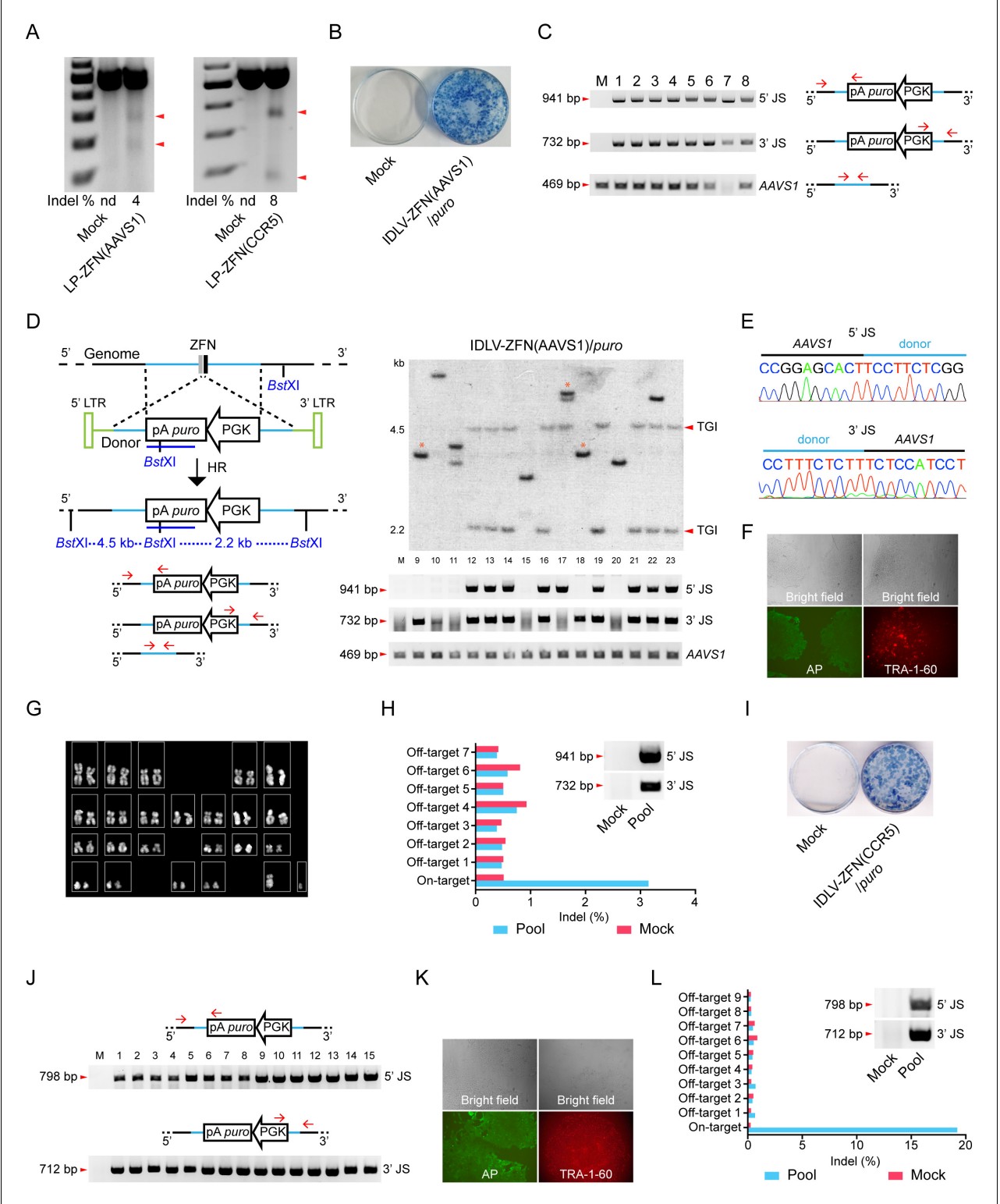

**Figure 3.** Targeted gene integration in iPSCs by IDLV-ZFN(AAVS1)/*puro* and IDLV-ZFN(CCR5)/*puro*. (**A**) Gene disruption mediated by transduction of LP-ZFN(AAVS1) and LP-ZFN(CCR5) in iPSCs. (**B**) Puromycin-resistant iPSC clones appearing after transduction with IDLV-ZFN(AAVS1)/*puro*. Clones were stained with methylene blue. (**C**) PCR analyses of targeted gene integration by IDLV-ZFN(AAVS1)/*puro* in clones 1–8. Primer sets used for amplification of 5' and 3' junction sites were the same as in *Figure 1F*, except that amplification of the *AAVS1* locus without the insertion was included as a control. (**D**) Southern blot (top) and PCR analyses (bottom) of targeted gene integration in clones 9–23 resulting from transduction with IDLV-ZFN(AAVS1)/*puro*. Schematic illustration of Southern blot mimics that in *Figure 1K*, except that a longer *puro* probe was used. Fragments indicated with * are indicative

*Figure 3 continued on next page*

*Figure 3 continued*

of imprecise TGI involving NHEJ at either the 5' or 3' junction site. (E) Representative sequence chromatograms of amplified 5' and 3' junctions (see also *Supplementary file 1A*). (F) Alkaline phosphatase (left) and TRA-1-60 (right) staining of a representative iPSC clone derived from IDLV-ZFN (AAVS1)/*puro* treatment. (G) Karyotyping analysis of a representative iPSC clone generated after transduction with IDLV-ZFN(AAVS1)/*puro*. (H) Indel frequencies obtained by deep sequencing of on- and top-ranked off-target sites in pools of iPSC clones generated by transduction with IDLV-ZFN (AAVS1)/*puro*. Inserted PCR analysis confirms the presence of TGI in a pool of iPSCs consisting of approximately 110 iPSC clones. (I) Puromycin-resistant iPSC clones appearing after transduction with IDLV-ZFN(CCR5)/*puro*. Clones were stained with methylene blue. (J) PCR analyses of targeted gene integration in iPSCs (clones 1–15) obtained by transduction with IDLV-ZFN(CCR5)/*puro*. (J) Alkaline phosphatase (left) and TRA-1-60 (right) staining of a representative iPSC clone derived from IDLV-ZFN(CCR5)/*puro* treatment. (L) Indel frequencies obtained by deep sequencing of on- and top-ranked off-target sites in pools of iPSC clones generated by transduction with IDLV-ZFN(CCR5)/*puro*. Inserted PCR analysis confirms the presence of TGI in a pool of iPSCs consisting of approximately 105 iPSC clones.

The following figure supplement is available for figure 3:

**Figure supplement 1.** Deep sequencing analysis of cleavage frequencies at the on-target and top ranking off-target loci of *AAVS1* (A) and *CCR5* (B) in pools of iPSC clones.

In conclusion, lentivirus-based co-delivery of nuclease proteins and a HR donor represents a powerful and safe approach for inserting transgenes into predetermined 'safe harbors' in the human genome.

## Discussion

On average, four out of five lentiviral integrations land in genes (*Mitchell et al., 2004*). It is still unclear to which extent this biased integration pattern induces toxicity due to insertional mutagenesis, but new approaches to alter the integration properties of lentiviral vectors remain attractive. Different strategies have been employed to reprogram the integration machinery of lentiviral vectors including (i) addition of a DNA-binding zinc-finger protein to the integrase (*Bushman and Miller, 1997*), (ii) fusion of a chromosomal binding domain to LEDGF, a cellular co-factor of the integrase (*Gijsbers et al., 2010*), and (iii) insertion of lentiviral DNA intermediates by transposases and recombinases (*Vink et al., 2009*; *Staunstrup et al., 2009*; *Moldt et al., 2011*; *Cai et al., 2014b*). These approaches successfully alter the integration preference, but do not allow site-specific insertion into a preferred locus. Here, we present an alternative homology-directed insertion strategy facilitated by nuclease-directed DNA cleavage. By co-packaging designer nucleases and donor sequences into lentiviral vectors and utilizing cellular HR repair pathways to integrate the transgene, we can guide lentiviral cargo toward a safe harbor in up to 84% of HEK293T cells and 89% of iPSCs that were transduced with the 'all-in-one' vector system. The efficiency may likely be further improved by exploiting heterodimer-forming rather than homodimer-forming FokI domains (*Doyon et al., 2011*).

TGI into the *AAVS1* and *CCR5* loci of stem cells has previously been achieved using HR-based strategies exploiting two- or three-component systems based on delivery of plasmids, IDLVs, and baculovirus vectors as well as combinations of mRNA encoding ZFNs and IDLVs serving as the HR donor (*Lombardo et al., 2011*; *Genovese et al., 2014*; *Tay et al., 2013*; *Tiyaboonchai et al., 2014*; *Hockemeyer et al., 2009*; *Lei et al., 2011*). Despite the potential of genetic therapies based on TGI in stem cells, like CD34$^+$ cells and iPSCs, existing technologies remain insufficient and highly toxic (*Genovese et al., 2014*; *Hoban et al., 2015*; *Merling et al., 2015*). Exploiting the capacity of lentiviruses to co-deliver nucleases and a transgene cassette, we showed TGI in CD34$^+$ cells transduced with ZFN-loaded IDLVs without affecting the viability, proliferation, or colony-forming capacity of the cells. However, the efficiency was only modest, suggesting that a selective growth advantage of the TGI-engineered cells, as in SCID-X1, would be required for successful engraftment. Possibly, efficacy could be boosted by combining virus treatment with exposure to chemicals enhancing stem cell self-renewal, by supplying NHEJ inhibitors (*Chu et al., 2015*), or by manipulating cell cycle conditions in favor of HR (*Marniemi et al., 1975*).

Stable transduction rates with ZFN-loaded IDLVs leading to TGI are undoubtedly reduced relative to conventional lentiviral gene delivery strategies. In iPSCs, however, transduction with IDLVs carrying ZFNs targeting either the *AAVS1* or the *CCR5* locus generated a high number of puromycin-resistant clones. Remarkably, 34 out of a total of 38 analyzed clones contained the transgene

expression cassette inserted precisely into the locus targeted by the ZFNs. Also, as typically observed in relation to the standard use of IDLVs, we found that a few clones contained an additional, randomly inserted transgene cassette, which could be of relevance for potential future therapeutic use. Importantly, however, we found that lentiviral ZFN protein delivery did not lead to detectable off-target cleavage. Hence, cleavage in a total of 16 predicted off-target loci (7 for *AAVS1* and 9 for *CCR5*) did not appear as analyzed by deep sequencing of large pools of clones carrying the transgene cassette. If desired, this approach can be combined with removal of the selection marker by seamless excision, exploiting for example lentiviral protein transduction of *piggyBac* transposase (*Cai et al., 2014b*). Remarkably, in a large pool of iPSCs, most if not all carrying a site-targeted gene insertion in the *CCR5* locus, off-target cleavage and indel formation at the neighboring *CCR2* locus was not evident after deep sequencing analysis. This is in contrast to higher levels of *CCR2* targeting reported in previous studies using other delivery strategies based on adenoviral vectors, plasmid transfection, or administration of recombinant proteins (*Perez et al., 2008*; *Gaj et al., 2012*), suggesting that lentiviral delivery of site-targeted endonucleases supports safer engineering of the genome. Together, our findings demonstrate that lentivirus particles can ferry programmable nucleases, facilitating homology-driven, site-directed genomic insertion of a co-delivered transgene. Such co-delivery ensures time-limited exposure of the treated cell to nucleases and, yet, results in frequent TGI with particular applicability in iPSCs.

## Materials and methods

### Plasmid construction

To construct pLV/AAVS1-donor-fluc-PGK and pLV/AAVS1-donor-egfp-PGK, inserts were amplified from pLV/fluc-PGK-PBT (*Cai et al., 2014b*) and pLV/egfp-PGK-PBT (cloned similarly as pLV/fluc-PGK-PBT), respectively, using primer sets YJ219F-YJ210R (primers are listed in *Supplementary file 2A*). The inserts were subsequently digested by *BsiWI*/*Sal*I and cloned into pLV/AAVS1-donor-LS (*Cai et al., 2014a*) that was digested with the same pair of enzymes. pLV/AAVS1-donor-puro-PGK and pLV/CCR5-donor-puro-PGK were cloned similarly by amplifying inserts from pLV/puro-PGK-PBT using primer set YJ262F-YJ210R and ligated with pLV/AAVS1-donor-LS and pLV/CCR5-donor-LS (*Cai et al., 2014b*), respectively. Restriction enzymes used for inserts and vectors were *BsrGI*/*Sal*I and *BsiWI*/*Sal*I, respectively. All donor sequences are provided in *Supplementary file 3*. The packaging construct, pMDlg/pRRE-D64V-218, for production of IDLVs was constructed by introducing a D64V mutation into the integrase of pMDlg/pRRE by overlap PCR. For the first round PCR, primers AgeI-int(s)-int-D64V(as) and int-D64V(s)-AccIII-int(as) were used, respectively, with pMDlg/pRRE as the templates. For the second round PCR, primers AgeI-int(s) and AccIII-int(as) were used with the PCR product from the first round as template. The inserts and vector pMDlg/pRRE were then both digested by *Age*I and *Acc*III before ligation.

### Cells and culture conditions

HEK293 and HEK293T were cultured in Dulbecco's Modified Eagle's Medium (Lonza, Basel, Switzerland). Culture medium was supplemented with 10% fetal bovine serum, 100 U/ml penicillin, 100 μg/ml streptomycin and 250 μg/ml L-glutamine. Human cord blood was obtained from Aarhus University Hospital. CD34[+] cells were purified from umbilical cord blood using EasySep Human Cord Blood CD34 Positive Selection Kit (STEMCELL Technologies, Grenoble, France) and cultured in StemSpan serum-free expansion medium (STEMCELL Technologies) supplemented with human early-acting cytokines (PeproTech, Rocky Hill, NJ) including stem cell factor (SCF) 100 ng/ml, Flt3 ligand (Flt3-L) 100 ng/ml, thrombopoietin (TPO) 50 ng/ml, and interleukin 3 (IL-3) 20 ng/ml and enhancer of self-renewal UM171 50 nM (STEMCELL Technologies). Induced pluripotent stem cells, described in a previous publication (*Kang et al., 2015*), were cultured using TeSR-E8 medium (STEMCELL Technologies) on Vitronectin XF (STEMCELL Technologies) coated culture dish. All cells were cultured at 37°C and 5% (vol/vol) $CO_2$.

### Lentiviral vector production and titration

In this work, 'LP' (as in LP-ZFN(AAVS1) and LP-ZFN(CCR5)) was used to designate lentiviral particles which did not carry a vector genome but contained a pair of ZFN proteins targeting the *AAVS1* and

*CCR5* locus, respectively. When a lentiviral genome was included, the vectors were referred to as 'IDLVs'. IDLV-ZFN(AAVS1)/*egfp*, IDLV-ZFN(AAVS1)/*puro* and IDLV-ZFN(AAVS1)/*fluc* all contained a pair of *AAVS1*-targeting ZFNs as well as RNAs, which upon reverse transcription formed DNA donors for homologous recombination at the *AAVS1* locus. The 'donor' genes carried by these vectors were *egfp*, *puro* and *firefly luciferase*, respectively, all driven by the PGK promoter. Preparations of LPs and IDLVs were produced as previously described (*Cai et al., 2014a*). Briefly, LP-ZFN(AAVS1) and LP-ZFN(CCR5) were produced by co-transfection of pMD.2G, pZFNL(*AAVS1* or *CCR5*)-PH-gag-pol-D64V and pZFNR(*AAVS1* or *CCR5*)-PH-gagpol-D64V. IDLV/*egfp*, IDLV/*puro* and IDLV/*fluc* were produced by co-transfection pMD.2G, pRSV-Rev, pMDlg/pRRE-D64V-218, and donor-containing vector (either pLV/AAVS1-donor-egfp-PGK, pLV/AAVS1-donor-puro-PGK or pLV/AAVS1-donor-fluc-PGK). IDLV-ZFN(AAVS1)/*egfp*, IDLV-ZFN(AAVS1)/*puro*, IDLV-ZFN(AAVS1)/*fluc* and IDLV-ZFN(CCR5)/*puro* were produced by co-transfection pMD.2G, pRSV-Rev, pMDlg/pRRE-D64V-218, pZFNL (AAVS1)-PH-gagpol-D64V (or pZFNL(CCR5)-PH-gagpol-D64V), pZFNR(AAVS1)-PH-gagpol-D64V (or pZFNR(CCR5)-PH-gagpol-D64V), and donor-containing vector (either pLV/AAVS1-donor-egfp-PGK, pLV/AAVS1-donor-puro-PGK, pLV/AAVS1-donor-fluc-PGK, or pLV/CCR5-donor-puro-PGK). Concentrations of HIV-1 p24 were measured by ELISA assay (XpressBio, Frederick, MD) according to the manufacturer's protocol. The MOIs of IDLV-ZFN(AAVS1)/*egfp* and IDLV/*egfp* were determined by virus transduction to HEK293T cells followed by flow cytometry analysis allowing quantification of the percentage of eGFP$^+$ cells three days after transduction.

## Confocal microscopy

For confocal microscopy, HEK293T cells or HEK293 cells were seeded at a concentration of $2\times10^5$ cells/well 1 day before transfection or transduction. HEK293T cells were co-transfected with 0.75 μg pMD.2G and 1.3 μg pHA-ZFNL(gfp)-PH-gagpol-D64V (*Cai et al., 2014a*) and 1.3 μg pHA-ZFNR (gfp)-PH-gagpol-D64V (*Cai et al., 2014a*) or with 0.75 μg pMD.2G and 2.6 μg pGFP-PH-gagpol-D64V (*Cai et al., 2014a*). Cells were fixed after 24 hr. HEK293 cells were transduced with LP-HA-ZFN(gfp) and incubated at 4°C for 1 hr to synchronize virus entry. The cells were then incubated at 37°C for 1 hr, 12 hr, 24 hr and 48 hr, respectively. To detect lentiviral particles, LP-HA-ZFN(gfp) and LP-eGFP were attached to poly-L-lysine coated coverslips by spinoculation at 1200 g for 99 min. In all the experiments, cells or lentiviral particles were washed three times with PBS before fixation with 4% paraformaldehyde. Coverslips were stored in 70% ethanol for at least 15 min. To visualize HA-tagged ZFNs, fixed cells or lentiviral particles were incubated with mouse-derived primary HA mono-clonal antibody (Covance, Princeton, NJ) and secondary Alexa Fluor 488 Donkey anti-Mouse IgG (LifeTechnologies). To visualize p24, the lentiviral particles were stained with rabbit-derived primary p24 polyclonal antibody (ThermoFisher Scientific, Waltman, MA) and secondary Alexa Fluor 568 Donkey-anti-Rabbit IgG (LifeTechnologies). The nuclei were stained by DAPI. Sequential imaging was done by using a 488 nm line of a multiline argon laser (detection of Alexa-488), the 405 nm diode laser (detection of DAPI) and 561 nm DPSS laser (detection of Alexa-568) on a confocal microscope (LSM 710, ZEISS) using 63 × oil immersion objective with a numerical aperture of 1.4.

## Lentiviral vector transduction in CD34$^+$ cells and iPSCs

In protocol 1, CD34$^+$ cells were transduced with IDLV-ZFN(AAVS1)/*egfp* or IDLV/*egfp* at MOI = 5. In protocol 2, CD34$^+$ cells were pretreated with 300 ng p24 Vpx-loaded lentiviral particles for 6 hr and then transduced with IDLV-ZFN(AAVS1)/*egfp* at MOI = 20. On the next day, cells were transduced with 300 ng p24 LP-ZFN(*AAVS1*). Both protocols used EF-C nanofibrils to boost virus transduction to CD34$^+$ cells. The iPSCs were pretreated with 2 μM ROCK inhibitor (Sigma-Aldrich, St. Louis, MO) for 1 hr and dissociated into single cells in the presence of ROCK inhibitor before transduced with LP-ZFN(CCR5), LP-ZFN(AAVS1), IDLV-ZFN(AAVS1)/*puro* and IDLV-ZFN(CCR5)/*puro* at a dose of 500 ng p24 per $0.5\times10^5$ cells. Five days after transduction, iPSCs treated with IDLV-ZFN(AAVS1)/*puro* and IDLV-ZFN(CCR5)/*puro* were selected with puromycin (0.5 μg/ml). Representative clones were further stained using alkaline phosphatase and TRA-1-60 live stain dyes (Life Technologies). For colony-forming unit assays, after transduction with and incubation for 4 days, 1200 CD34$^+$ cells were mixed with Human Methylcellulose Complete Media (R&D Systems Inc., Minneapolis, MN) and transferred to a 35-mm plate. The plates were incubated in a humidified atmosphere with 5% (vol/vol) $CO^2$ at 37°C for two weeks.

## Flow cytometry

HEK293T cells were seeded in 6-well plates ($1 \times 10^5$ cells/well) one day before transduction with 500 ng p24 IDLV-ZFN(AAVS1)/*egfp* or IDLV/*egfp*. Cells were fixed with 4% paraformaldehyde. Data were collected on a FACSCalibur (BD Biosciences, Franklin Lakes, NJ). CD34[+] Cells were analyzed on LSRFortessa cell analyzer (BD Biosciences) for viability using the LIVE/DEAD fixable near-IR dead cell stain kit (Life Technologies) 4 and 9 days after transduction and for the presence of eGFP[+] cells 9 days after transduction.

## Luciferase assay

HEK293T cells were seeded in 6-well plates ($1 \times 10^5$ cells/well) one day before transduction with 500 ng p24 IDLV-ZFN(AAVS1)/*fluc* or IDLV/*fluc*. Luminescence analysis was carried out 3, 10 and 18 days after transduction, respectively. $0.5 \times 10^5$ cells were transferred to 96-well plates 6 hr before luciferase activity analysis using ONE-Glo luciferase assay system (Promega, Madison, WI). Luminescence analysis was performed on a multi-sample plate-reading luminometer (Berthold Technologies, Bad Wildbad, Germany).

## Surveyor nuclease assay

Surveyor nuclease assay (Integrated DNA Technologies, Leuven, Belgium) was used to determine the frequencies of gene disruption as described previously (*Cai et al., 2014a*). Briefly, iPSCs or CD34[+] cells were incubated with ZFN-loaded LPs for 48 hr and collected for DNA extraction. HEK293T cells transduced by IDLV-ZFN(AAVS1)/*egfp* or IDLV-ZFN(AAVS1)/*fluc* were also subjected to Surveyor nuclease assay. *CCR5* and *AAVS1* fragments were amplified using primer sets YJ224F-YJ225R and YJ222F-YJ223R, respectively. The PCR products were incubated with 1 μl Surveyor nuclease plus 1 μl enhancer at 42°C for 1 hr after denaturation and re-annealing. The cleavage products were separated on a 1.5% agarose gel and stained with ethidium bromide. The percentage of indels was determined by the formula $100 \times (1-(1-\text{fraction cleaved})^{1/2})$, wherein the fraction cleaved is the sum of the cleavage product peaks divided by the sum of the cleavage product and parent peaks.

## Southern blot analysis

HEK293T cells were transduced with IDLV-ZFN(AAVS1)/*puro* or IDLV/*puro* at a dose of 500 ng p24 per $10^5$ cells and selected by puromycin. The puromycin-resistant clones were isolated and expanded for Southern blot analysis. Fifteen micrograms of genomic DNA from each clone was digested overnight with *Bst*XI or *Eco*NI (for subsequent hybridization with a *puromycin* and an *AAVS1* probe, respectively) before gel electrophoresis and vacuum blotting. The *puromycin* fragment was derived from pLV/AAVS1-donor-puro-PGK using *Bst*XI and *Sal*I, whereas the *AAVS1* probe was amplified from genomic DNA by nested PCR using primer sets YJ599F-YJ600R and YJ601F-YJ602R, successively. The iPSC clones derived from treatment with IDLV-ZFN(AAVS1)/*puro* were prepared and analyzed in the same way, except for the use of a puromycin probe amplified from pLV/AAVS1-donor-puro-PGK using primers Puro-1 and Puro-2. The probes were randomly labeled using the Prime-It random primer labeling kit (Agilent Technologies, Santa Clara, CA) according to the manufacturer's instructions.

## PCR analyses of targeted gene integration

Nested PCR reactions were performed to confirm homology-directed integration at the *AAVS1* and *CCR5* loci. In the case of the *AAVS1* locus, primer sets YJ706F-0279 and YJ707F-YJ738R were used as first round and second round PCRs, respectively, to detect 5′ junctions. YJ251F-YJ205R and YJ252F-YJ666R primer sets were used as first round and second round primer set, respectively, to detect 3′ junctions. For insertion into the CCR5 locus, YJ825F-0279 and YJ833F-YJ738R were used for amplification of 5′ junctions. Primer sets YJ251F-YJ253R and YJ252F-YJ208R were used to detect 3′ junctions. The *CCR5* locus was amplified using primer set YJ224F-YJ225R, whereas the *AAVS1* locus was amplified using primer set YJ222F-YJ223R.

## Karyotyping

For karyotyping analysis, actively growing cells on 6-well plate were treated with KaryoMAX Colce-midSolution (0.1 µg/ml, ThermoFisher Scientific) for 2 hr at 37°C. Cells were then swollen with warm 0.56% KCL for 10–15 min at 37°C. After centrifugation, the cell pellets were fixed with fresh cold fixative solution (methanol: acetic acid, 3:1) for 30 min on ice. Cells were gently washed twice with fixative solution. A drop of fixed cell suspension was dropped on a slide and then dried in a humid condition. Metaphase chromosomes were imaged by microscope and analyzed by Quips CGH software.

## Next generation sequencing

The on-target and top ranking off-target loci of *AAVS1* and *CCR5* predicted by PROGNOS were amplified from genomic DNA of iPSCs treated with IDLV-ZFN(AAVS1)/*puro* and IDLV-ZFN(CCR5)/*puro* as well as from non-treated iPSCs using Phusion High-Fidelity PCR Master Mix (ThermoFisher Scientific). Primers and predicted off-target loci are listed in *Supplementary file 2B*. The purified PCR fragments amplified from each locus were mixed in equimolar amounts. Adaptors were ligated to the pools, which were then deep sequenced with an Illumina HiSeq2500, HiSeq Rapid Run 125 bp PE mode (GATC Biotech, Constance, Germany). The indel frequencies were analyzed using CRISP-Resso (*Pinello, 2015*) using the CRISPRessoPooled amplicons mode, while ignoring substitutions. Alternatively, and for further validation, the HiSeq reads were trimmed with Sickle (*Joshi and Fass, 2011*), merged with FLASH (*Magoč and Salzberg, 2011*), and mapped to the amplicons using bow-tie2 (*Langmead and Salzberg, 2012*). Reads with an average Phred score below 23 were discarded. The reads were then aligned using EMBOSS water (*Rice, 2000*) and analyzed for indels. The maximum-likelihood estimate of the true-indel fraction was calculated as previously described (*Hsu et al., 2013*).

## Acknowledgements

The authors are grateful to Haiqing Tang (Department of Clinical Medicine, Aarhus University Hospital, Denmark) for assistance related to the use of human cord blood and to Trine Skov Petersen (Department of Biomedicine, Aarhus University, Denmark) for additional technical assistance. Also, the authors would like to thank to Jan Münch (Institute of Molecular Virology, Ulm University Medical Center, Germany) for providing EF-C peptides.

# Additional information

### Funding

| Funder | Grant reference number | Author |
|---|---|---|
| Danish Council for Independent Research, DFF Medical Sciences | DFF-4004-00220 | Jacob Giehm Mikkelsen |
| The Hørslev Foundation | | Jacob Giehm Mikkelsen |
| Civilingeniør Frode V. Nyegaard og Hustrus Fond | | Jacob Giehm Mikkelsen |
| Aase og Ejnar Danielsens Fond | | Jacob Giehm Mikkelsen |

The funders had no role in study design, data collection and interpretation, or the decision to submit the work for publication.

### Author contributions

YC, Conception and design, Acquisition of data, Analysis and interpretation of data, Drafting or revising the article; AL, YZ, CS, MVA, Revising article, Final approval, Acquisition of data, Analysis and interpretation of data; SL, Revising article, Final approval, Analysis and interpretation of data; NU, Revising article, Final approval, Contributed unpublished essential data or reagents; YL, MRJ, Revising article, Final approval, Analysis and interpretation of data, Contributed unpublished

essential data or reagents; JGM, Conception and design, Analysis and interpretation of data, Drafting or revising the article

## Author ORCIDs
Yonglun Luo, http://orcid.org/0000-0002-0007-7759
Jacob Giehm Mikkelsen, http://orcid.org/0000-0002-1322-3209

## Additional files

### Supplementary files
• Supplementary file 1. (A) Sanger sequencing of donor-genome junctions and the *AAVS1* locus. The PCR products of 5' junction and 3' junction sites as well as the *AAVS1* locus from 8 analyzed iPSC clones were Sanger sequenced. The black letters represent genomic DNA at the *AAVS1* locus, whereas blue letters represent donor-derived sequences. WT indicates wild-type sequence; indel (+1bp) indicates 1-bp insertion; N.A., not available. (B) Sanger sequencing of PROGNOS-predicted off-target site 1, 2 and 3. The blue sequence indicates the predicted binding site of the right ZFN; the green sequence indicates predicted binding site of the left ZFN; letters highlighted in red indicate mismatches.

• Supplementary file 2. (A) Primers used for molecular cloning and PCR amplification of junction sites. (B) Primers used to amplify PROGNOS-predicted off-target sites.

• Supplementary file 3. Donor sequences. Homology arms and transgene expression cassettes in pLV/AAVS1-donor-egfp-PGK, pLV/AAVS1-donor-fluc-PGK, pLV/AAVS1-donor-puro-PGK and pLV/CCR5-donor-puro-PGK are provided. For each transgene expression cassette, sequences are annotated as listed below the sequence and indicated with different colors.

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
