## [Decision Letter]

Thank you for submitting your work entitled "Targeted, homology-driven gene insertion in stem cells by ZFN-loaded 'all-in-one' lentiviral vectors" for consideration by *eLife*. Your article has been reviewed by two peer reviewers, and the evaluation has been overseen by Charles Sawyers as the Senior Editor.

The reviewers have discussed the reviews with one another and the Reviewing editor has drafted this decision to help you prepare a revised submission.

Your manuscript presents a method for targeted gene integration (TGI) into human cells via an all-in-one integration defective lentiviral system coupled with ZFN protein and donor template delivery. Previously, your group demonstrated the efficacy of this approach in human cell lines, and has now extended this work into human cord blood and human iPS cells.

Overall, the manuscript is well written and clearly presented. The experimental objectives and methods are well described, and the key findings should benefit those interested in similar therapeutic gene-delivery strategies. However, there are a few key points that impact the novelty of the results. Furthermore, some of the finding implications would challenge ZFN-loaded 'all-in-one' vectors as an efficient method for targeted gene integration (TGI). These concepts are addressed below.

It is essential that your revision, if you choose to resubmit, address the following two points, which are elaborated in further detail in the specific comments by the referees.

1) Do you have evidence supporting the use of the all-in-one system at a second locus, beyond *AAVS*?

2) Can you elaborate more on the issue of non-specific integration, either by inclusion of additional data to address this point or by more detailed acknowledgement of the issue?

We also request that you address the additional issues surrounding methodology and data presentation that are raised by both referees.

*Reviewer #1:*

The manuscript by Cai and colleagues presents a method for targeted gene integration (TGI) into human cells via an all-in-one integration defective lentiviral system coupled with ZFN protein and donor template delivery. Previously, this group demonstrated the efficacy of this approach in human cell lines, and has now extended this work into human cord blood and human iPS cells. In general, I found the study very well described, however there is some lack of clarity in experimental descriptions, figure legends and methods.

1) It is unclear to me whether the study demonstrates replication at a second independent target site from *AAVS1*. I cannot find in the text nor the legend which site they are targeting in Figure 2 They compared EF-C and RetroNectin using the *AAVS1* target, but the PCR in Figure 2 shows CCR5 this the target? I see in Figure 1 that *CCR5* is amplified in 1F and 1I, why would you not assay *AAVS1* to see if the target site is intact? This lack of clarity and description is a problem throughout the text and figures as the authors have oversimplified the reagent descriptions. For example, instead of just saying IDLV-ZFN/*egfp*, it should read IDLV-CCR5/*egfp* or IDLV-AAVS1/*egfp* in each part, or something similar. The manuscript needs a thorough review in this regard and Figures and legends should be described and labeled accordingly. If Figure 2 is also showing targeting *AAVS1*, then it seems they have the tools available to attempt the TGI into the *CCR5* locus in either the CD34^+^ or iPS cells.

2) I am not familiar with RetroNectin or EF-C. Please define and reference these approaches.

3) In the last paragraph of the Results, it is unclear what is meant by sequencing the *AAVS1* locus. Do the authors mean sequencing the non-modified allele?

4) Please describe the methods used to identify the off target sites reported. Was off target analysis performed with *CCR5*, such as looking at *CCR2* in targeted clones?

5) The authors should describe and provide the full sequences (or reference) the donor vector homology arms. How long are they? I also cannot seem to discern the actual target sequences for the ZFN reagents for *AAVS1* or *CCR5*. Are the constructs used in this study available upon request?

6) Figure 1 – is there some explanation of the varying intensity of the TGI band on the Southern? Is it possible that some TGI clones have more than 1 copy inserted? I am not clear if the restriction site chosen would distinguish whether multiple tandem head-to-tail copies are inserted in some clones.

*Reviewer #1 (Additional data files and statistical comments):*

It would be great if the authors could provide full sequence maps of these vectors, they are likely to be widely sought after and it would facilitate others trying to replicate this and similar approaches.

*Reviewer #2:*

The manuscript entitled, "Targeted, homology-driven gene insertion into stem cells by ZFN-loaded 'all-in-one' lentiviral vectors" by Cai et al. describes the use of homology-directed integration of transgenes into the 'safe-harbor' *AAVSI* locus by integrase-defective lentiviral vectors (IDLVs) that are co-packaged with zinc-finger nuclease (ZFN) proteins and a transgene-containing donor template. The authors demonstrate the specificity of targeting to the *AAVSI* locus in HEK293 cells by conventional PCR and Southern blot analyses. They show that greater long-term transgene expression (eGFP, Luciferase, or puromycin) is achieved when cells are transduced with IDLV packaged with ZFN (targeted integration, 2.5%) than without (random integration, >1%). Interestingly, the authors note that integrations were not wholly by homologous recombination (HR), but a few events were due to non-homologous end-joining (NHEJ). They further apply the strategy in CD34^+^ hematopoietic stem cells and iPSCs, and show successful transgene integration into the *AAVS1* locus. Finally, the authors argue that treatment with IDLV-ZFN/*puro* vectors do not alter the integrity of iPSC stemness evidenced by stem cell marker expression and karyotyping.

Overall, the manuscript is well written and clearly presented. The experimental objectives and methods are well described, and the key findings should benefit those interested in similar therapeutic gene-delivery strategies. However, there are a few key points that impact the novelty of the results. Furthermore, some of the finding implications would challenge ZFN-loaded 'all-in-one' vectors as an efficient method for targeted gene integration (TGI). These concepts are addressed below.

As pointed out in the text, the work described in the manuscript is a follow-up study to the paper in *eLife* (2014) entitled: "Targeted genome editing by lentiviral protein transduction of zinc-finger and TAL-effector nucleases" by the same group. In this previous work, the authors described the use of similar 'all-in-one' lentiviral vectors co-packaged with ZFN or TALEN proteins to promote HR-mediated gene editing at the *CCN5* and *AAVS1* loci. The two main differences between the previous work, and the current one is that, 1) the authors attempt to explore in depth, the specificity and efficacy by which 'all-in-one' lentiviral vectors can promote TGI; and 2), the authors describe a protocol for TGI in hematopoietic stem cells and iPSCs.

1) There are a handful of published reports that have explored the insertion of transgenes (including ZNFs, albeit in a two-vector strategy) into the *AAVS1* locus in ES cells and iPSCs and these should be cited:

Select publications:

Tay FC, et al., Targeted transgene insertion into the AAVS1 locus driven by baculoviral vector-mediated zinc finger nuclease expression in human-induced pluripotent stem cells. J Gene Med. 2013 Oct; 15(10):384-95. doi: 10.1002/jgm.2745.

Tiyaboonchai, A et al., Utilization of the AAVS1 safe harbor locus for hematopoietic specific transgene expression and gene knockdown in human ES cells. Stem Cell Res. 2014 May; 12(3):630-7. doi: 10.1016/j.scr.2014.02.004. Epub 2014 Feb 21.

2) The authors do a satisfactory job describing the insertion events occurring at the *AAVS1* site. However, their evidence of site-specific integration falls short. As the authors have illustrated in Figure 1 and Figure 1—figure supplement 1, low frequency, off-target integration events do occur. However, it is in this reviewer's opinion that conventional PCR and Southern blot analysis are limited in their ability to detect the prevalence of additional off-target events. Although the authors do query 5 top off-target sites, these are only predicted sites that are based on the designed ZFN's position weight matrix using the PROGNOS tool. They do not consider off-target integration events at hotspots that are independent of ZFN activity. For instance, the use of genome-wide approaches have found that translocations via non-specific NHEJ events due to CRISPR/Cas9- or TALEN-mediated gene editing are numerous:

Frock RL et al., Genome-wide detection of DNA double-stranded breaks induced by engineered nucleases. Nat Biotechnol. 2015 Feb;33(2):179-86. doi: 10.1038/nbt.3101. Epub 2014 Dec 15.

This is especially critical when the authors demonstrate that 5-8% of the insertions are due to NHEJ (Figure 1). Therefore, the frequency by which delivered ZFNs will promote non-specific NHEJ events on a genome-wide scale is unclear. The authors do attempt to address whether lentiviral treatment can cause genomic instability by demonstrating stem-cell integrity and karyotyping in Figure 3. However, only a single-representative clone was analyzed as described in the last paragraph of the Results (Figure 3). Furthermore, Figure 3 is of low quality and karyotyping to assess translocation events should be performed using spectral-karyotyping, on more than one clone.

3) In addition, Figure 1, lane 9 suggests that the *AAVS1* locus has been successfully targeted by IDLV-ZFN/*puro*, however a second integration event is detectable. This observation suggests that the methodology can lead to multiple integration events in the genome, within single clones. If this is indeed the case, the authors need to discuss how this can impact the efficacy of TGI via IDLV-ZFN/transgene vectors in therapeutic settings.

In conclusion, since the study appears to parallel many aspects of the research group's previous work, the unique findings are limited in range, and may not be highly influential to the field, this reviewer cannot recommend the manuscript by Cai et al. to be accepted for publication in *eLife*.

[Editors' note: further revisions were requested prior to acceptance, as described below.]

Thank you for resubmitting your work entitled "Targeted, homology-driven gene insertion in stem cells by ZFN-loaded 'all-in-one' lentiviral vectors" for further consideration at *eLife*. Your revised article has been favorably evaluated by Charles Sawyers as the Senior editor and two reviewers.

The manuscript has been improved but there are two remaining issues that need to be addressed before acceptance, as outlined below:

1) Please acknowledge any published reports that have explored the insertion of transgenes into the *AAVS1* locus in ES cells and iPSCs using ZNFs using 2-vector systems, per the comment below:

"It should again be noted that the study's results expand upon the group's previous innovation, and a handful of published reports have already explored the insertion of transgenes into the *AAVS1* locus in ES cells and iPSCs using ZNFs (but using 2-vector systems)."

2) Please address the concern raised about the quality of the blot in Figure 1, as discussed below:

"Figure 1 blot is now of very low quality. It seems like the same figure used in the previous version of the manuscript. Please ensure that this is fixed."

---

## [Author Response]

[…] Overall, the manuscript is well written and clearly presented. The experimental objectives and methods are well described, and the key findings should benefit those interested in similar therapeutic gene-delivery strategies. However, there are a few key points that impact the novelty of the results. Furthermore, some of the finding implications would challenge ZFN-loaded 'all-in-one' vectors as an efficient method for targeted gene integration (TGI). These concepts are addressed below.

It is essential that your revision, if you choose to resubmit, address the following two points, which are elaborated in further detail in the specific comments by the referees.

1) Do you have evidence supporting the use of the all-in-one system at a second locus, beyond AAVS?

2) Can you elaborate more on the issue of non-specific integration, either by inclusion of additional data to address this point or by more detailed acknowledgement of the issue?

To address the two major points listed in the decision letter, we have carried out substantial additional experimentation. Hence, focusing on targeted gene insertion in iPSCs, we generated ZFN-loaded ‘all-in-one’ lentiviral vectors targeting a second locus (the human *CCR5* locus) and verified site-directed gene insertion also into this locus. Notably, in all of 15 analyzed iPSC clones the transgene cassette was inserted into the *CCR5* locus. Also, in the revision we are the including the analysis of 15 additional iPSC clones treated with AAVS1-directed ‘all-in-one’ vectors bringing the total number of analyzed *AAVS1* clones up to 23. Of these 23 clones, 19 harbored an inserted transgene in the *AAVS1* locus. Moreover, to address the second point, we have now markedly improved our analysis of specificity and potential off-target cleavage. Hence, we performed next-generation sequencing on pools of iPSC clones appearing after treatment with *AAVS1* and *CCR5*-directed ‘all-in-one’ vectors. Importantly, deep sequencing of panels of top-ranked off-target gave no indication of cleavage at off-target sites. Encouragingly, our analysis also showed that targeted gene insertion into the *CCR5* locus by lentiviral ZFN protein delivery was not accompanied by cleavage and indel formation in the *CCR2* locus, a well-known off-target locus, which is known from the literature to be frequently targeted when other delivery approaches are utilized.

*We also request that you address the additional issues surrounding methodology and data presentation that are raised by both referees.*

*Reviewer #1:*

*The manuscript by Cai and colleagues presents a method for targeted gene integration (TGI) into human cells via an all-in-one integration defective lentiviral system coupled with ZFN protein and donor template delivery. Previously, this group demonstrated the efficacy of this approach in human cell lines, and have now extended this work into human cord blood and human iPS cells. In general, I found the study very well described, however there is some lack of clarity in experimental descriptions, figure legends and methods.*

*1) It is unclear to me whether the study demonstrates replication at a second independent target site from AAVS1. I cannot find in the text nor the legend which site they are targeting in Figure 2 They compared EF-C and RetroNectin using the AAVS1 target, but the PCR in Figure 2 shows CCR5 this the target? I see in Figure 1 that CCR5 is amplified in 1F and 1I, why would you not assay AAVS1 to see if the target site is intact? This lack of clarity and description is a problem throughout the text and figures as the authors have oversimplified the reagent descriptions. For example, instead of just saying IDLV-ZFN/egfp, it should read IDLV-CCR5/egfp or IDLV-AAVS1/egfp in each part, or something similar. The manuscript needs a thorough review in this regard and Figures and legends should be described and labeled accordingly. If Figure 2 is also showing targeting AAVS1, then it seems they have the tools available to attempt the TGI into the CCR5 locus in either the CD34^+^ or iPS cells.*

We agree with the reviewer that the names and descriptions of the reagents were perhaps oversimplified in the original submission. In the original manuscript we only targeted the*AAVS1* locus, and this is why we decided to simplify names of lentiviral particles and viral vectors. PCR amplification of the *CCR5* locus was only occasionally used as a control, and this clearly led to some confusion. Things are different in the revised manuscript, as we have now included vectors that target the human *CCR5* locus. This also means that the nomenclature has been changed (and improved). Throughout the text, we now use the format ‘LP-ZFN(locus)’ to describe lentiviral particles loaded with ZFNs targeting the mentioned locus and ‘IDLV-ZFN(locus)/*gene*’ to describe IDLVs loaded with locus-directed ZFNs and carrying a specific transgene (*egfp, fluc* or *puro*). The nomenclature is explained where relevant (e.g. Results, first paragraph) and is used throughout the manuscript and in figures.

2) I am not familiar with RetroNectin or EF-C. Please define and reference these approaches.

We have revised the text (Results, sixth paragraph), which now includes further information on both RetroNectin and EF-C.

*3) In the last paragraph of the Results, it is unclear what is meant by sequencing the AAVS1 locus. Do the authors mean sequencing the non-modified allele?*

This part of the manuscript has been substantially revised, and the particular sentence, which now appears in the ninth paragraph of the Results, has been changed. The reviewer is correct that we sequenced the *AAVS1* site to search for modifications in the allele without the insertion. To clarify this point the wording is now as follows: “Interestingly, sequencing of the *AAVS1* locus provided evidence of biallelic cleavage in at least one of the clones (clone 5), resulting in a 1-bp insertion in the allele that did not contain the transgene ([Supplementary-material SD1-data]).”

4) Please describe the methods used to identify the off target sites reported. Was off target analysis performed with CCR5, such as looking at CCR2 in targeted clones?

Originally, we only looked for off-target cleavage in the three top-ranked off-target sites in the 8 iPSC clones carrying an insertion in the *AAVS1* locus. We did not address *CCR5*-directed insertion in the original manuscript and therefore did not get into the issue of off-target insertion in the *CCR2* locus. However, as mentioned, we have extended the analysis now to include lentiviral vectors loaded with *CCR5*-targeting ZFNs and verify the use of such IDLVs for prominent site-directed transgene insertion into the *CCR5* locus. Notably, all clones generated by this approach harbored the site-directed gene insertion (new Figure 3). We obviously agree with both reviewers that a NGS-based approach was required to strengthen the analysis. This analysis has now been carried out on lentivirally transduced iPSCs harboring insertions in either the *AAVS1* or the *CCR5* locus. Hence, the analysis focuses on identifying cleavage leading to indels in the non-modified on-target allele or in panels of top-ranked off-target sites, including in the case of *CCR5* the *CCR2* locus (off-target 3). NGS data showing on-target cleavage in both the *AAVS1* and *CCR5* loci (3 and 19%, respectively) do not provide evidence of any off-target cleavage in any of the predicted sites (new Figure 3). Interestingly, targeting of the *CCR2* after delivery of CCR5-directed ZFN proteins is at the background level, indicating that cells harboring an on-target *CCR5* insertion do not have indels in the most prominent and well-known off-target locus. Hence, in comparison to other delivery methods protein delivery using lentiviral vectors seems gentler on the off-target loci even in cells where the nucleases are cutting both on-target alleles.

As noted above, we have also extended the analysis of *AAVS1*-targeted IDLVs including now more a thorough analysis of a total of 23 iPSC clones. Southern blot data on 15 of these clones have now been added. As a result, this part of the manuscript has been fully revised. The description of *AAVS1*-directed gene insertion in iPSCs is now in the ninth and tenth paragraphs of the Results. Following this section, we have included new work related to targeting of the *CCR5* locus (Results, eleventh paragraph). Both these sections include data based on NGS. Issues related to the absence of *CCR2* targeting are now also included in the Discussion (last paragraph).

5) The authors should describe and provide the full sequences (or reference) the donor vector homology arms. How long are they? I also cannot seem to discern the actual target sequences for the ZFN reagents for AAVS1 or CCR5. Are the constructs used in this study available upon request?

Yes, we did not include information of these issues in the first place. All information is now provided in [Supplementary-material SD3-data].

6) Figure 1 – is there some explanation of the varying intensity of the TGI band on the Southern? is it possible that some TGI clones have more than 1 copy inserted? I am not clear if the restriction site chosen would distinguish whether multiple tandem head-to-tail copies are inserted in some clones.

No, we have no evidence (based on specific band patterns) suggesting that any of these integrations are tandem insertions. It is clear that other types of insertions, besides site-targeted insertion, may occur. These are most likely random insertions of IDLV DNA intermediates, but may also represent homology-independent insertions in the targeted locus (like in clones 5 and 13 in Figure 1). In accordance with other Southern data (Figure 1—figure supplement 1), we believe that the varying band intensities represent differences in the amounts of DNA.

*Reviewer #1 (Additional data files and statistical comments):*

*It would be great if the authors could provide full sequence maps of these vectors, they are likely to be widely sought after and it would facilitate others trying to replicate this and similar approaches.*

Sequences of the relevant parts of all donor constructs (homology arms, promoter, gene) are now available in [Supplementary-material SD3-data]. Obviously, all constructs and plasmid maps will be available upon request.

Reviewer #2:

[…] Overall, the manuscript is well written and clearly presented. The experimental objectives and methods are well described, and the key findings should benefit those interested in similar therapeutic gene-delivery strategies. However, there are a few key points that impact the novelty of the results. Furthermore, some of the finding implications would challenge ZFN-loaded 'all-in-one' vectors as an efficient method for targeted gene integration (TGI). These concepts are addressed below.

As pointed out in the text, the work described in the manuscript is a follow-up study to the paper in eLife (2014) entitled: "Targeted genome editing by lentiviral protein transduction of zinc-finger and TAL-effector nucleases" by the same group. In this previous work, the authors described the use of similar 'all-in-one' lentiviral vectors co-packaged with ZFN or TALEN proteins to promote HR-mediated gene editing at the CCN5 and AAVS1 loci. The two main differences between the previous work, and the current one is that, 1) the authors attempt to explore in depth, the specificity and efficacy by which 'all-in-one' lentiviral vectors can promote TGI; and 2), the authors describe a protocol for TGI in hematopoietic stem cells and iPSCs.

1) There are a handful of published reports that have explored the insertion of transgenes (including ZNFs, albeit in a two-vector strategy) into the AAVS1 locus in ES cells and iPSCs and these should be cited:

Select publications:

Tay FC, et al., Targeted transgene insertion into the AAVS1 locus driven by baculoviral vector-mediated zinc finger nuclease expression in human-induced pluripotent stem cells. J Gene Med. 2013 Oct;15(10):384-95. doi: 10.1002/jgm.2745.

Tiyaboonchai, A et al., Utilization of the AAVS1 safe harbor locus for hematopoietic specific transgene expression and gene knockdown in human ES cells. Stem Cell Res. 2014 May;12(3):630-7. doi: 10.1016/j.scr.2014.02.004. Epub 2014 Feb 21.

It has been a major priority for us to keep the Introduction short, but the mentioned references are certainly relevant, and we have now cited both references in the Introduction (first paragraph).

2) The authors do a satisfactory job describing the insertion events occurring at the AAVS1 site. However, their evidence of site-specific integration falls short. As the authors have illustrated in Figure 1 and Figure 1—figure supplement 1, low frequency, off-target integration events do occur. However, it is in this reviewer's opinion that conventional PCR and Southern blot analysis are limited in their ability to detect the prevalence of additional off-target events. Although the authors do query 5 top off-target sites, these are only predicted sites that are based on the designed ZFN's position weight matrix using the PROGNOS tool. They do not consider off-target integration events at hotspots that are independent of ZFN activity. For instance, the use of genome-wide approaches have found that translocations via non-specific NHEJ events due to CRISPR/Cas9- or TALEN-mediated gene editing are numerous:

Frock RL et al., Genome-wide detection of DNA double-stranded breaks induced by engineered nucleases. Nat Biotechnol. 2015 Feb;33(2):179-86. doi: 10.1038/nbt.3101. Epub 2014 Dec 15.

This is especially critical when the authors demonstrate that 5-8% of the insertions are due to NHEJ (Figure 1). Therefore, the frequency by which delivered ZFNs will promote non-specific NHEJ events on a genome-wide scale is unclear. The authors do attempt to address whether lentiviral treatment can cause genomic instability by demonstrating stem-cell integrity and karyotyping in Figure 3. However, only a single-representative clone was analyzed as described in the last paragraph of the Results (Figure 3). Furthermore, Figure 3 is of low quality and karyotyping to assess translocation events should be performed using spectral-karyotyping, on more than one clone.

We agree with the reviewer that the evidence of site-directed gene insertion should not fall short. We acknowledge the point that more information on insertion patterns (than we originally provided) would support the claims made in the original submission. We chose to focus our additional analyses on iPSCs, in which we find frequent site-directed gene insertion using the ‘all-in-one’ gene delivery system. We carried out PCR of 5’ and 3’ junction sites as well as Southern blot analysis on 15 additional clones generated with *AAVS1*-directed vectors. As shown in Figure 3, the PCR data were in complete accordance with the Southern data clearly identifying clones that contained a site-directed gene insertion. Obviously, Southern blotting additionally provides information on the appearance of additional insertions in specific clones. One example is clone 22 in Figure 3. Hence, in this context Southern blotting is sufficient to identify events of randomly inserted lentiviral vectors. It is well known from the literature that even IDLVs (integrase-defective lentiviral vectors) may occasionally insert into the genome, and this is obviously confirmed by our data. However, it would not be relevant here to carry out genome-wide studies to identify such off-target random insertion sites that are not necessarily related to ZFN function. Although it cannot be excluded that such events may occur, it is much more relevant, as noted by the reviewer, to focus on the analysis on indel formation in off-target recognition sites. One can argue that a complete genome-wide analysis would be relevant for this matter (also to detect translocations), but we believe that an extensive, NGS-based analysis of an extended panel of predicted off-target loci was sufficient to demonstrate the specificity of the approach. In addition to work involving *AAVS1*-targeted vector, we also (as mentioned) included a whole new series of experiments including *CCR5*-targeted vectors. Here, we confirmed site-directed insertion by PCR (Figure 3) and concentrated on the NGS analysis of ZFN activity at a panel of 9 off-target loci including the *CCR2* locus (Figure 3). The lack of detectable cleavage in the *CCR2* locus (as shown by the lack of indel formation in this locus in a large pool of clones with ZFN-induced *CCR5*-directed insertions and evidence of substantial biallelic cleavage) is to us a strong indication that this ZFN delivery approach – possibly due to the strongly time-restricted presence of ZFNs – has unprecedented specificity and is safe. Overall, we believe that we have carefully addressed key issues related to achieving site-directed gene insertion. We acknowledge the importance of the genomic integrity and long-term potential of ZFN-engineered iPSCs, but also feel that such studies belong in a future report that may not fit in this ‘Research Advances’ format.

The reviewer alludes to experiments shown in Figure 1, arguing that between 5 and 8% of the insertions are due to NHEJ (Figure 1). It should be pointed out that the mentioned assays indicate rates of imprecise repair by NHEJ in non-targeted alleles and, thus, do not quantify insertions involving NHEJ.

3) In addition, Figure 1, lane 9 suggests that the AAVS1 locus has been successfully targeted by IDLV-ZFN/puro, however a second integration event is detectable. This observation suggests that the methodology can lead to multiple integration events in the genome, within single clones. If this is indeed the case, the authors need to discuss how this can impact the efficacy of TGI via IDLV-ZFN/transgene vectors in therapeutic settings.

The reviewer is correct; transduction with IDLVs may potentially lead to insertion, and this is an often-overlooked fact in the literature. Even with a relatively potent site-directed gene insertion strategy (as the one we demonstrate here), we occasionally observed such secondary events, which may potentially affect potential therapeutic use. To stress that such events occur and may be of relevance also for therapeutic applications, we have added a single sentence in the Discussion: “Also, as typically observed in relation to the standard use of IDLVs, we found that a few clones contained an additional, randomly inserted transgene cassette, which could be of relevance for potential future therapeutic use.”

[Editors' note: further revisions were requested prior to acceptance, as described below.]

The manuscript has been improved but there are two remaining issues that need to be addressed before acceptance, as outlined below:

1) Please acknowledge any published reports that have explored the insertion of transgenes into the AAVS1 locus in ES cells and iPSCs using ZNFs using 2-vector systems, per the comment below:

"It should again be noted that the study's results expand upon the group's previous innovation, and a handful of published reports have already explored the insertion of transgenes into the AAVS1 locus in ES cells and iPSCs using ZNFs (but using 2-vector systems)."

We certainly agree that our work expands upon our previous findings and the attempts and success of other groups to target gene insertion using nuclease-based strategies. To acknowledge these previous reports, we have added the following to the Discussion: “TGI into the *AAVS1* and *CCR5* loci of stem cells has previously been achieved using HR-based strategies exploiting two- or three component systems based on delivery of plasmids, IDLVs, and baculovirus vectors as well as combinations of mRNA encoding ZFNs and IDLVs serving as the HR donor 3, 4, 7, 8, 21, 22.”

2) Please address the concern raised about the quality of the blot in Figure 1, as discussed below:

"Figure 1 blot is now of very low quality. It seems like the same figure used in the previous version of the manuscript. Please ensure that this is fixed."

It is correct that the quality of Figure 1 in the previous submission was suboptimal. However, the resolution of the figure that we originally uploaded was good, and the figure was somehow formatted during the submission process (apparently during formation of the manuscript PDF). We are uploading the original figure again and are hoping, thus, that the problem is fixed. We checked the separate figure file (from our last submission) and found that this figure was OK, so we expect this to be a formatting problem, which is now hopefully fixed.